

# Bayesian multistate models for measuring invasive carp movement and evaluating telemetry array performance

Jessica C. Stanton[1], Marybeth K. Brey[1], Alison A. Coulter[2], David R. Stewart[3] and Brent Knights[1]

[1] Upper Midwest Environmental Sciences Center, United States Geological Survey, La Crosse, WI, United States of America
[2] Department of Natural Resource Management, South Dakota State University, Brookings, SD, United States of America
[3] Division of Biological Sciences, United States Fish and Wildlife Service, Albuquerque, NM, United States of America

## ABSTRACT

Understanding the movement patterns of an invasive species can be a powerful tool in designing effective management and control strategies. Here, we used a Bayesian multistate model to investigate the movement of two invasive carp species, silver carp (*Hypophthalmichthys molitrix*) and bighead carp (*H. nobilis*), using acoustic telemetry. The invaded portions of the Illinois and Des Plaines Rivers, USA, are a high priority management zone in the broader efforts to combat the spread of invasive carps from reaching the Laurentian Great Lakes. Our main objective was to characterize the rates of upstream and downstream movements by carps between river pools that are maintained by navigation lock and dam structures. However, we also aimed to evaluate the efficacy of the available telemetry infrastructure to monitor carp movements through this system. We found that, on a monthly basis, most individuals of both species remained within their current river pools: averaging 76.2% of silver carp and 75.5% of bighead carp. Conversely, a smaller proportion of silver carp, averaging 14.2%, and bighead carp, averaging 13.9%, moved to downstream river pools. Movements towards upstream pools were the least likely for both species, with silver carp at an average of 6.7% and bighead carp at 7.9%. The highest probabilities for upstream movements were for fish originating from the three most downstream river pools, where most of the population recruitment occurs. However, our evaluation of the telemetry array's effectiveness indicated low probability to detect fish in this portion of the river. We provide insights to enhance the placement and use of these monitoring tools, aiming to deepen our comprehension of these species' movement patterns in the system.

# INTRODUCTION

Successful control of invasive species is reported less often in aquatic systems than terrestrial systems (*Simberloff, 2021*) and full eradication of aquatic invaders is rarely successful (*Simberloff, 2014*). When a decision is made to attempt either an eradication or

Corresponding author
Jessica C. Stanton, jcstanton@usgs.gov

management of a biological invader, effort and resources are allocated to three main areas of activity. The first involves the identification of the primary ecological and/or evolutionary processes that are driving or facilitating the invasion (*Gurevitch et al., 2011*). The second focuses on the design and testing of control strategies that disrupt the key processes driving the invasion (*Buhle, Margolis & Ruesink, 2005*). The third area centers on the allocation of resources to implement the management actions. The execution of these activities does not always occur sequentially, as it is sometimes necessary to take immediate action to contain or minimize the damage before fully understanding the systematic drivers or developing optimal long-term strategies. Therefore, adaptive management and changes in priorities often occur as new or additional information is acquired (*Williams & Brown, 2016*).

An example of this scenario is currently playing out with the spread of the invasive silver carp (*Hypophthalmichthys molitrix*) and bighead carp (*H. nobilis*), collectively known as bigheaded carps, in the rivers of the United States (*Chick & Pegg, 2001*). Since the mid-1970s, state and federal natural resource management agencies have endeavored to control these two species, which have caused both significant economic (*Pimentel, Zuniga & Morrison, 2005*; *Spacapan, Besek & Sass, 2016*) and ecological (*Solomon et al., 2016*; *Chick et al., 2020*; *Altenritter et al., 2022*) damage in the waters where they have invaded. Bigheaded carp invasions result in changes to plankton and zooplankton species composition and abundance (*DeBoer, Anderson & Casper, 2018*) and are linked to declines in some native fish species (*Fletcher et al., 2019*; *Chick et al., 2020*). In addition to declines in native fish populations, the nuisance silver carp jumping behavior poses to recreational water users and anglers can negatively impact local economies (*Wittmann et al., 2015*).

An issue of on-going concern is preventing the bigheaded carp invasion from reaching the Laurentian Great Lakes (*Cudmore et al., 2012*; *Chapman et al., 2021*). The level of harm caused by these species in the areas where they have already established, and the risk of further damage should they reach the Great Lakes, has prompted collaborations across multiple entities and agencies. These groups are working together to deploy a multi-modal strategy to control further spread while simultaneously researching the biological drivers behind the invasion. Insights from this research aim to pave the way for innovative control methods (*Cupp et al., 2021*). New management planning tools, such as the Spatially Explicit Invasive Carp Population (SEICarP) model, are now available to guide the strategic deployment of these control methods, ensuring both effectiveness and efficiency (*Cupp et al., 2021*; *ICRCC, 2022*; *Kallis et al., 2023*).

The most likely path for bigheaded carps to reach the Laurentian Great Lakes is through the Illinois Waterway (IWW). This extensive waterway stretches from the mouth of the Chicago River at Lake Michigan to the confluence of the Illinois and Mississippi Rivers (*Bandy, 1996*). Water flow and navigation through the IWW is controlled through a series of locks and dams, which create a series of navigation pools (*i.e.,* the segments of the river between successive locks and dams, hereafter referred to as 'pools'). Generally, the population density of bigheaded carps in the IWW decreases moving upstream from the confluence with the Mississippi River towards the Great Lakes. The current invasion front in the IWW is stable at approximately 76 km downstream from Lake Michigan, potentially because of intensive ongoing management (*ICRCC, 2023*). Most of

the population recruitment in the IWW is likely occurring in the three most downstream pools on the Illinois River (*Garvey et al., 2015*; *Norman & Whitledge, 2015*). Although spawning and small numbers of eggs and larvae have been observed in the upper pools closer to the invasion front (*Parkos et al., 2023*), ongoing monitoring has not detected evidence that this has resulted in successful recruitment to juvenile or adult stages. This implies that most or all adult bigheaded carps present in the three most upstream pools made upstream movements from one of the three lower reach pools (*Zhu et al., 2018*). Further understanding the pool-to-pool movement probabilities of adult carp has been identified as a key information need to the ongoing management and control program (*Garvey et al., 2015*; *Coulter et al., 2018*; *ICRCC, 2022*; *Kallis et al., 2023*).

Acoustic telemetry has been highly useful to assess the movement dynamics of bigheaded carps in this system (*Coulter et al., 2018*). This approach involves surgically implanting bighead carp, silver carp, and their surrogate fish species (*e.g.*, common carp, *Cyprinus carpio*) with uniquely coded acoustic transmitters (tags) that are detected on an array of 69 kHz acoustic receivers strategically deployed throughout the system. These telemetry data can be used to estimate movement probabilities among river pools (*Lubejko et al., 2017*; *Coulter et al., 2018*), demographic parameters such as survival or fishing mortality (*Lees et al., 2021*), and can be used in conjunction with mark-recovery or tag return data (*Bacheler et al., 2009*). These estimates can help parameterize management planning tools such as SEICarP (*Cupp et al., 2021*; *ICRCC, 2022*; *Kallis et al., 2023*). Additionally, acoustic telemetry data can be used to assess potential bigheaded carp spawning movements, identify optimal spawning conditions, and inform other monitoring efforts (*e.g.*, for eggs or larvae). Furthermore, telemetry data can also guide additional management actions such as where and when to concentrate removal efforts or place dispersal barriers.

A recent analysis of bigheaded carp metapopulation dynamics using the SEICarp model in the IWW explored the possible effectiveness of various management strategies (*Kallis et al., 2023*). The movement-related parameters for this model were derived from the top models identified in *Coulter et al. (2018)* re-estimated using Markov chain Monte Carlo (MCMC). These movement-related parameters were estimated using acoustic telemetry data from 2012 to 2015. While *Kallis et al. (2023)* offered valuable insights into managing these bigheaded carp in the IWW, they also noted some behaviors of the model that were strongly influenced by the movement estimates used to parameterize the model and emphasized the need for updated estimates.

Here we provide updated estimates of pool-to-pool bigheaded carp movements in the IWW. Our multistate model is constructed similarly to the one in *Coulter et al. (2018)* (the model from which SEICarp parameters were based) but differs in two key areas. First, we used an expanded dataset, encompassing additional years of fish tag data (2012–2019) and additional fish as well as an expanded telemetry receiver array. Our dataset was assembled from university, state, and federal partners who were maintaining receivers or acoustically tagging fish within portions of the IWW during this period. Leveraging these partnerships, we have expanded the amount of available information included in the model, such as detections from receivers deployed for research projects which may have been unrelated to invasive carp, but that detect any fish tagged with a compatible transmitter. Second,

we modified how the model estimates the probability of fish detection to be based on the number of available receivers. This approach allowed us to examine the relative performance of the acoustic telemetry array for each pool and highlights opportunities to optimize on-going fish-tagging and receiver maintenance operations. Therefore, the primary goals of this study were twofold: (1) to provide updated pool-to-pool transition probabilities for bigheaded carps in the IWW and (2) to assess the IWW's acoustic telemetry array's performance. This assessment provides guidance for future planning and coordination among the management partnership for on-going monitoring work.

## METHODS

### Study site

Coverage of the acoustic telemetry array includes a portion of the IWW, extending from the confluence of the Illinois and Mississippi Rivers to the upstream extent of the bigheaded carp invasion front below Brandon Road Lock and Dam (Fig. 1). This section of the IWW is located in the State of Illinois, USA, and is composed of the Illinois River and a downstream portion of the Des Plaines River. It is divided into six pools, named for the downstream dam that forms each pool, except Alton which flows uninhibited into the Mississippi River. Each lock and dam structure is used to control water levels and maintain a minimum 2.74 m navigation channel throughout the river (Bhowmik et al., 2001). Each lock consists of a navigation lock and a dam, although the dam type and operations may vary. For the purposes of this study, we refer to the three most upstream pools–Dresden Island, Marseilles, and Starved Rock–as the 'upper reach,' while the three lower pools–Peoria, La Grange, and Alton–are referred to as the 'lower reach.'

Movement of fish between pools is partially restricted by these lock and dam structures, especially during low-flow river conditions (DeGrandchamp, Garvey & Colombo, 2008). The dam structures that separate the pools differ in their design, construction, and operation relative to river flow conditions and are described in more detail in Coulter et al. (2018). The three dams that form the pools of the upper reach are high-head gravity dams, mainly composed of Tainter gates that infrequently go into ''open-river'' conditions (i.e., where all gates are open, and fish can freely pass through those gates; (Montenero et al., 2018; Bouska et al., 2019). From 1985–2016 these upper reach dams were in open-river conditions for an average of 1.6 days/year (Montenero et al., 2018). Therefore, for most of the time at these dams, upstream fish passage is limited to the lock chamber. In contrast, the two dams in the lower reach (La Grange and Peoria) are composed mainly of wicket-style gates. When the wickets are lowered completely (i.e., to the riverbed) there are no impeding structures to water flow or fish movement. These lower reach dams went into open-river conditions an average of 143.2 days/year (Montenero et al., 2018).

### Telemetry data

Telemetry data for this project consist of three data streams: acoustic receiver deployment data, fish and transmitter (i.e., acoustic tag) deployment data, and detection data. We obtained these data through a shared multi-agency telemetry database maintained and

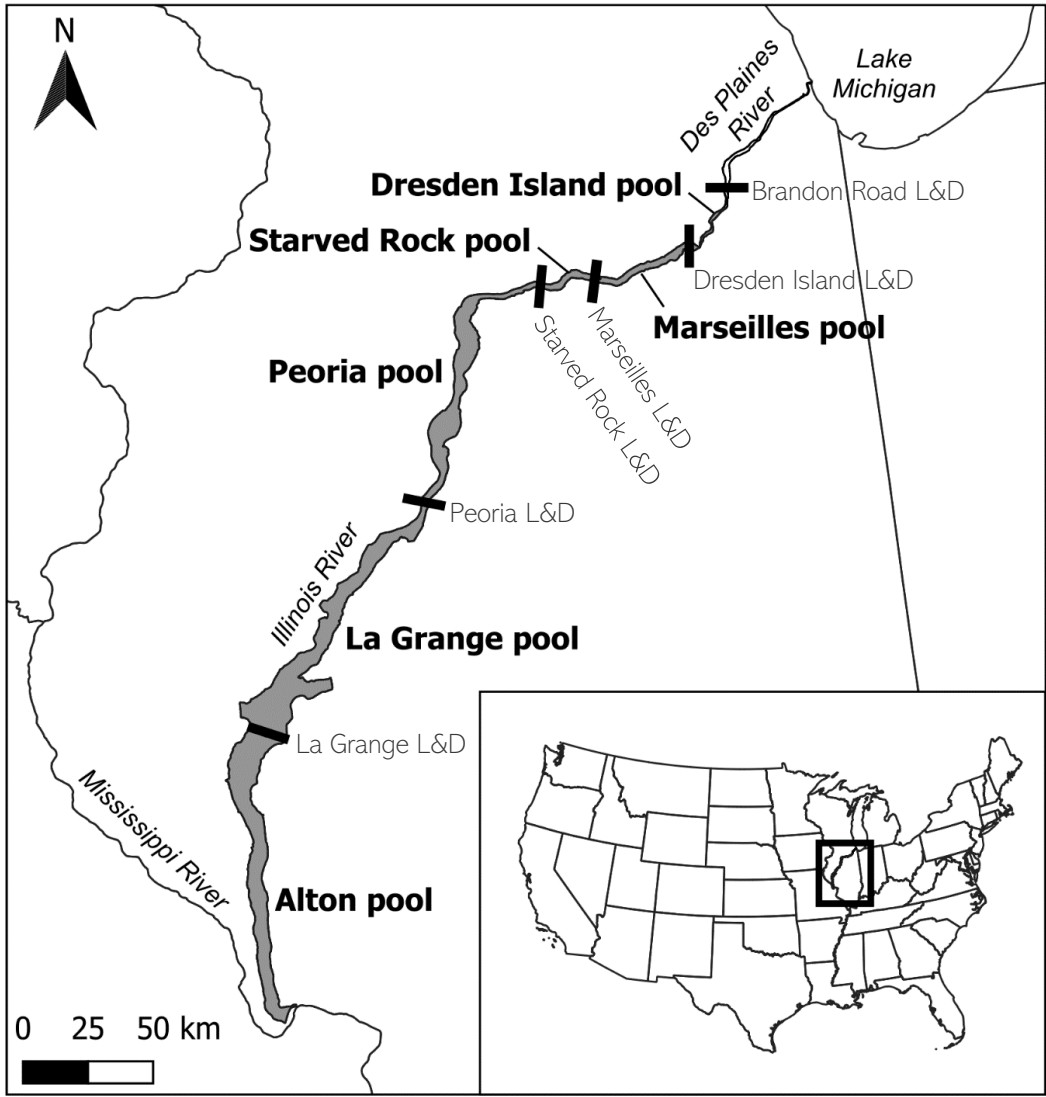

**Figure 1 Location of the study area on the lower portion of the Illinois Waterway (IWW) in the State of Illinois, USA.** The study area (shaded region) is composed of the Illinois River and the lower portion of the Des Plaines River below Brandon Road Lock and Dam. The study area is divided into six navigation pools, each named for the lock and dam (L&D) structure (black bars) that forms the downstream boundary, except for Alton pool, which flows uninhibited into the Mississippi River. For this study, Alton, La Grange, and Peoria pools are referred to as the 'lower reach', and Starved Rock, Marseilles, and Dresden Island pools compose the 'upper reach'. Map inset shows the study area relative to the contiguous United States. Tributaries, L&D structures beyond the boundaries of the study area, and details of the Chicago Area Waterway System are not shown (see *ICRCC, 2023* for history and additional details of the IWW). Map created with QGIS ver. 3.28.

quality-controlled by the US Geological Survey (FishTracks; https://umesc-gisdb03.er.usgs.gov/Fishtracks). All data owners or contributors agreed to the use of their data in this study.

The acoustic receiver array was composed of a series of Innovasea 69 kHz stationary receivers (VR2W, VR2Tx, or VR2C; Innovasea, Bedford, Nova Scotia, Canada) distributed

throughout the study area. This array was deployed and maintained by five agencies: Southern Illinois University–Carbondale (SIU), US Army Corps of Engineers (USACE) Chicago District, US Fish and Wildlife Service Carterville Fish and Wildlife Conservation Office (USFWS CFWCO) Wilmington Substation, Illinois Natural History Survey (INHS), and the US Geological Survey (USGS) Upper Midwest Environmental Sciences Center (UMESC). The number of deployed receivers varied among pools and within each pool over the duration of the study. At a minimum, receiver deployment data contained date and location a receiver was deployed, dates on which data were offloaded, and dates receivers were retrieved or lost (if known).

Acoustic tag data were obtained for fish captured and surgically implanted with acoustic transmitters in the IWW by partners. These data included information on fish species, fish length and occasionally weight, transmitter tag code, date tagged, expected tag expiration date, and location tagged. We used fish tagging information for silver and bighead carp from SIU, USACE, USFWS CFWCO Wilmington Substation, and Western Illinois University. Fish tagging procedures generally followed methods described in *Lubejko et al. (2017)* and *Coulter et al. (2018)*.

Acoustic receiver data were downloaded a minimum of two times per year, depending on river conditions and agency schedules. Data offloaded from the acoustic receivers contained the transmitter tag codes that were detected, decoded with a date and time stamp of each detection. Joining these three data streams provided us with the tagging and movement history for each fish.

After compiling the receiver deployment data from each agency, we calculated the mean number of receivers deployed per day for each month by summing the total number of receivers that were deployed in each pool each day and dividing by the number of days in the month. The mean number of receivers deployed per day ranged from 0 to 21 (mean = 7.05, standard deviation [SD] = 3.99; Fig. 2). The number of fish that were tagged and released also varied within each pool over the duration of the study. We removed fish from further analysis if the fish was suspected to have died or dropped their tag shortly after tagging. We removed fish if the only detections were within the first 30 days following tagging or if the tag continued to be detected but was stationary. A tag was determined to be stationary if at least two of the following conditions were met: the totality of detection data consisted of a single detection event where the tag code was continuously detected on a single receiver or simultaneously on multiple receivers in proximity; the maximum distance between all receivers with detections was less than 1,000 m; or the only pool of detection was the pool the fish was tagged and released in. This set of rules was specifically developed for this study, based on a visual examination of graphical depictions of individual fish detection histories. After removing fish that met these criteria, our analysis incorporated data from 353 silver carp and 170 bighead carp (Fig. 3).

Pool residency was determined based on the location of the receiver recording the tag detection. Detections on receivers located at a lock separating one pool from another were not used to determine pool residency because it is often not possible to determine from which pool the signal was emanating and simultaneous detections on multiple receivers around these structures was common. For a portion of the analyses (multistate models), we

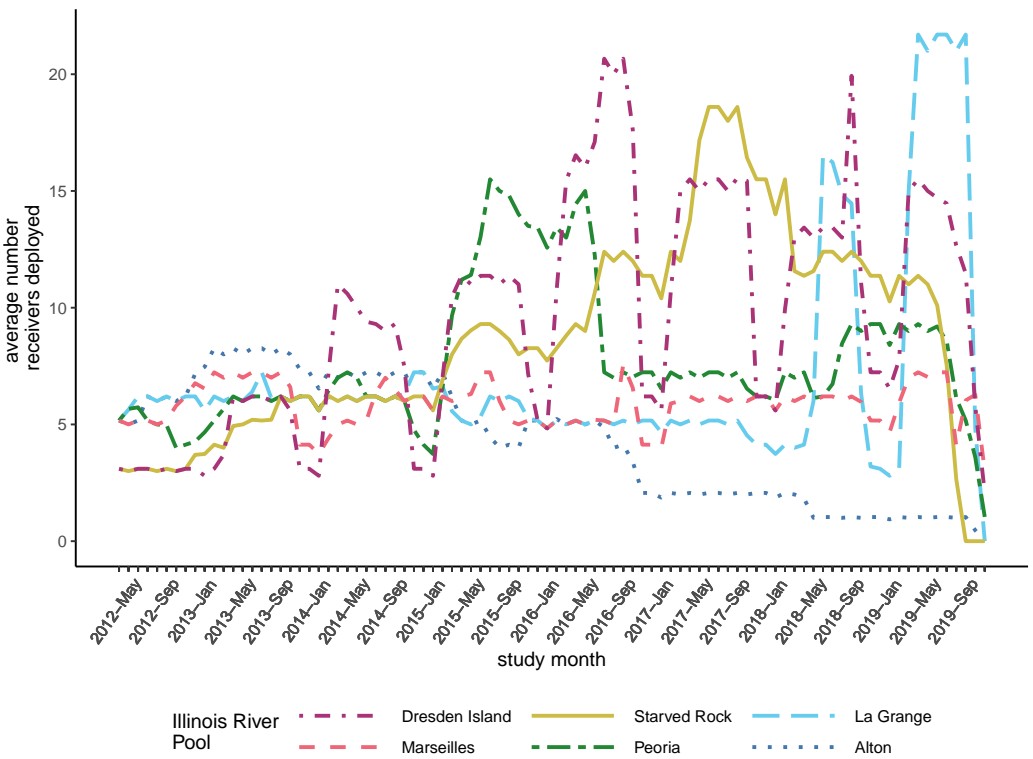

**Figure 2  Average number of receivers deployed per month in each of the Illinois River pools included in the model datasets.** Monthly average for each pool is calculated as the total number of receiver days divided by the number of days in the month. This dataset was assembled from contributed data from university, state, and federal partners maintaining receivers in these pools between 2012 and 2019 and include data from research projects of short duration and/or limited spatial coverage.

reduced detections to a monthly pool residency (*Coulter et al., 2018*), defined as the pool where the fish resided for the majority of a given month. For fish that were detected in more than one pool within a month, monthly residence pool was determined by the maximum number of days the fish was detected in each pool. In the case of a tie, residence pool was determined by the maximum number of detections recorded in each pool. Summarized data are available on ScienceBase (*Stanton et al., 2023*).

## Detection summary and analysis

To assess differences in the efficiency of the telemetry receiver array across the study system, we analyzed the detection patterns of fish based on which pool they were tagged and released in. We calculated the total number of days each fish had at least one detection (detection frequency) and the total duration of the detection period between the first detection and the last detection or tag expiration (detection period). We then used each measure as a response variable in a generalized linear mixed-effect model (GLMM) with release pool and species as predictor variables as well as the interaction between species and pool to account for differences between habitat use and receiver placement between pools. We also included a random effect for tagging year to control for possible between-year

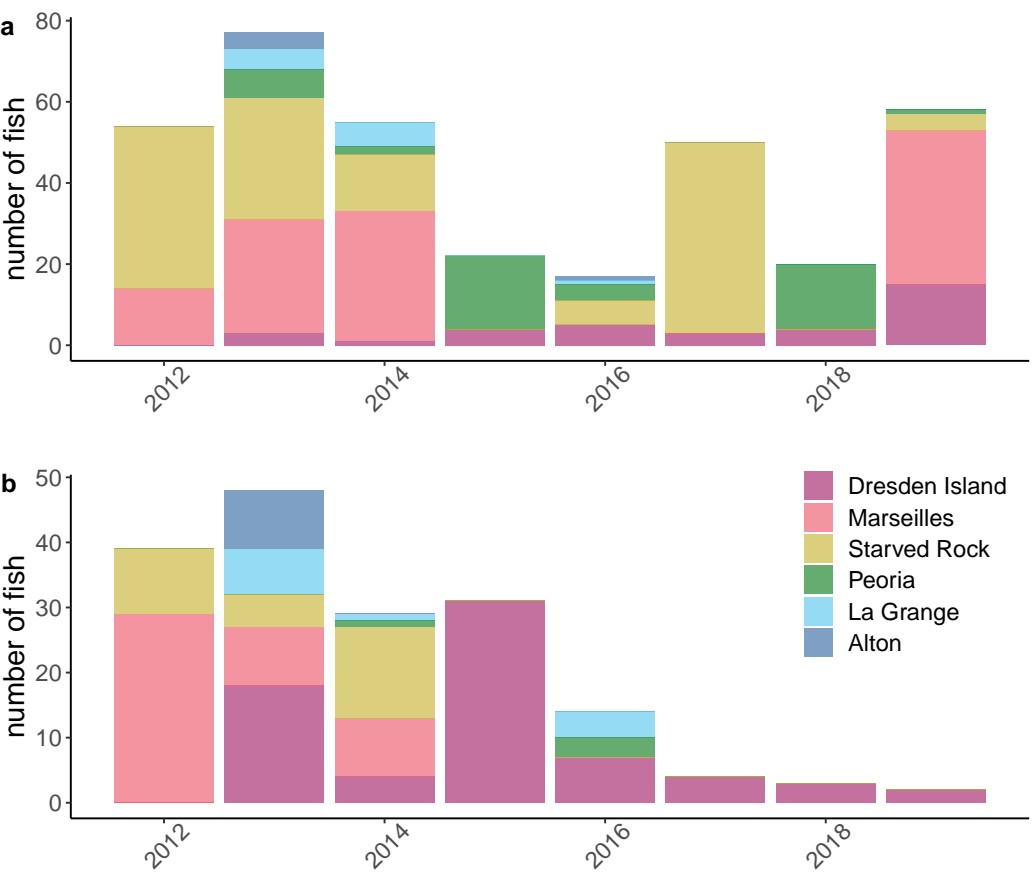

**Figure 3** **Numbers of silver carp (*Hypophthalmichthys molitrix*) and bighead carp (*H. nobilis*) included in the model datasets.** Counts of (A) silver carp and (B) bighead carp include only fish that passed data processing filters to remove suspected dead fish or dropped tags (see Methods) and are grouped by the Illinois River pool where fish were tagged and released each year. This dataset was assembled from contributed data from university, state, and federal partners acoustically tagging fish between 2012 and 2019 and include data from research projects of short duration and/or limited spatial coverage.

differences. We used a negative binomial distribution to model both response variables as they are over-dispersed count data. Models were fit using the lme4 package in R (*Bates et al., 2015*; *R Core Team, 2022*) with post-hoc pairwise comparisons between pools and species conducted using the emmeans package (*Lenth, 2022*). The denominator degrees of freedom were corrected using the Kenward-Roger approximation (*Kenward & Roger, 2009*).

## Multistate model

Reduced detection histories were used to construct a multistate model to characterize the movement of silver and bighead carp among pools in the IWW. The first step in parameterizing this model was to outline all the possible states the subjects could occupy and all the possible transitions between those states. We treated the study area as a closed system, where fish were either alive and occupying one of the six river pools or deceased. However,

as re-detection of tagged fish relies on operative implanted transmitters, we explicitly considered the state of the transmitter battery. Because environmental conditions, such as water temperature, can affect true battery life, we noted the approximate tag expiration date. Detections beyond the expected tag expiration date were not included in the analysis because signal rate and strength may not be as reliable as the battery near expiration, which can affect detection. Too few fish were reported as having been harvested to meaningfully contribute to the model, therefore we did not define that state.

Apart from pool occupancy, there were four possible fish statuses: 'alive with operable battery' (A/O), 'dead with operable battery' (D/O), 'alive with expired battery' (A/E), and 'dead with expired battery' (D/E). To account for these statuses, we constructed a state-transition matrix that determines the probability of staying in the same status or transitioning to another status from time $t$ (columns) to time $t + 1$ (rows) as:

|  | A/O | D/O | A/E | D/E |
|---|---|---|---|---|
| **A/O** | $\phi\delta$ | 0 | 0 | 0 |
| **D/O** | $(1-\phi)\delta$ | $\delta$ | 0 | 0 |
| **A/E** | $\phi(1-\delta)$ | 0 | $\phi$ | 0 |
| **D/E** | $(1-\phi)(1-\delta)$ | $(1-\delta)$ | $(1-\phi)$ | 1 |

where $\phi$ is the probability of survival, and $\delta$ is the probability of the battery remaining operable.

To model the pool-to-pool transitions, a sub-matrix, $\boldsymbol{M}$, was nested within the top left entry of the state-transition matrix ('alive, operable battery' status) to create the full complement of possible states. Sub-matrix $\boldsymbol{M}$ predicts the probability of living fish with operable batteries transitioning to pool $i$ from pool $j$ with probability $\psi_{i,j}$. Thus, the elements of $\boldsymbol{M}$ are $\phi \times \delta \times \psi_{i,j}$, and the state-transition matrix is expanded to include all pool-to-pool transitions for living fish with operable batteries that do not die or have their batteries expire before the next timestep. Column totals of the state-transition matrix sum to one.

Due to the small numbers of tagged fish in some pools for estimating pool-level survival probabilities, we assumed that probability of survival remained constant across all pools throughout the study. This assumption was consistent with the best-fit models from *Coulter et al. (2018)*. Although there is some evidence in the dataset and from other studies (*DeGrandchamp, Garvey & Colombo, 2008*; *Coulter et al., 2016*) that upstream movements may be correlated to seasonal factors, such instances in our dataset were rare relative to the number of additional parameters that would need to be incorporated into the model to explore seasonality. Therefore, we chose not to include time-variable probabilities of movement in our analysis.

For the model, we assumed that any fish, dead or alive, was in an 'expired battery' status once it reached its projected battery expiration date and asserted this in the data. We assert in the model that deceased fish with operational transmitters were not observable (*i.e.,* detected on a receiver). We acknowledge the possibility that fish that expire or expel their

transmitter within range of a receiver may continue to be detected. However, our data cleaning steps (see above) attempted to remove such stationary detections from the dataset. The observation matrix was:

| | A/O | D/O | A/E | D/E |
|---|---|---|---|---|
| **detected** | $\rho_{j,t}$ | 0 | 0 | 0 |
| **not detected** | $1-\rho_{j,t}$ | 1 | 0 | 0 |
| **battery expired** | 0 | 0 | 1 | 1 |

where $\rho_{j,t}$ is the probability of detecting a fish on any receiver deployed in pool $j$ in timestep $t$. Rows of the observation matrix descending from the top row describe the probability of being (1) detected on a receiver, (2) not detected, or (3) having an expired battery.

We expected the variation in the number of receivers deployed in each pool over the duration of the study to affect the probability that a fish with a functioning transmitter would be detected (*Melnychuk et al., 2017*; *Bunch et al., 2022*). To account for this in the model, we made the probability of detection a function of the mean receiver-days in each pool in each month. In the model the probability of detection $\rho_{j,t}$ for a given pool $j$ at time $t$, takes the form of a logistic function:

$$\rho_{j,t} = \frac{\exp(-5+\beta_j x_{j,t})}{1+\exp(-5+\beta_j x_{j,t})} \tag{1}$$

$$\beta_j \sim N(\theta,\sigma) \tag{2}$$

where $\beta_j$ isthe pool-level detection effect and is drawn from a normal distribution described by the overall mean effect, $\theta$, and standard deviation, $\sigma$. The mean number of receiver-days in pool $j$ in timestep $t$ is represented by $x_{j,t}$. If no receivers were deployed in a pool during a particular month, as may occur to protect equipment during winter months or if there were no active research projects, we would expect the probability of detection to be zero. For that reason, we set the intercept on the detection probability function as a constant, $-5$. This value forces the probability of detection curves to approach the origin (probability of detection approaches zero as the receiver deployment approaches zero) when the logit link is applied.

The state process is modeled as a Markov chain, where the true state at time $t+1$ is conditioned on the previous state at time $t$. This state transition process together with the observation process takes the form of a hidden Markov model. Model parameters were estimated using Markov chain Monte Carlo sampling in a Bayesian framework. Uniform distributions spanning from 0 to 1 were used as priors for $\phi$ and $\delta$; $\theta$ was drawn from a normal distribution with a mean of 0, and standard deviation of 1; and $\sigma$ was drawn from a uniform distribution between 0 and 5. We used Dirichlet priors for each set of pool-to-pool transition probabilities as it is the conjugate distribution to categorical distributions (*Frigyik, Kapila & Gupta, 2012*; *Turek, De Valpine & Paciorek, 2016*). The

 

**Table 1 Estimated marginal means and pair-wise comparisons between pools of detection frequency (number of days with detections).** Numbers on the diagonal in brackets are estimated marginal means for pools after accounting for random year effects. Numbers above the diagonal are *p*-values from *post hoc* pairwise comparisons between pools (values < 0.05 in bold).

| | Alton | La Grange | Peoria | Starved Rock | Marseilles | Dresden Island |
|---|---|---|---|---|---|---|
| **Silver Carp** | | | | | | |
| Alton | [15.8] | 1.000 | 0.9990 | **0.0003** | 0.0500 | **<0.0001** |
| La Grange | | [17.66] | 0.9981 | **<0.0001** | **0.0004** | **<0.0001** |
| Peoria | | | [24.58] | **<0.0001** | **<0.0001** | **<0.0001** |
| Starved Rock | | | | [133.96] | **0.0016** | 0.9874 |
| Marseilles | | | | | [72.03] | **0.0007** |
| Dresden Island | | | | | | [172.08] |
| **Bighead Carp** | | | | | | |
| Alton | [30.53] | 0.9663 | **0.0122** | 0.9124 | **0.0050** | **<0.0001** |
| La Grange | | [16.32] | 0.1298 | **0.0330** | **<0.0001** | **<0.0001** |
| Peoria | | | [2.72] | **<0.0001** | **<0.0001** | **<0.0001** |
| Starved Rock | | | | [55.71] | **0.0140** | **<0.0001** |
| Marseilles | | | | | [129.32] | 0.6994 |
| Dresden Island | | | | | | [197.28] |

model was fit using the nimble package in R (*De Valpine et al., 2017*; *De Valpine et al., 2022a*; *De Valpine et al., 2022b*). Models for both species were run with 3 chains with 10,000 iterations of burn in and 100,000 iterations for posterior estimation thinned by 1/3. Model convergence was determined by visual inspection of chains and confirmation that all top-level model parameters had Gelman and Rubin convergence statistics <1.1 using the coda package (*Plummer et al., 2006*). Code and data to run the models are available on ScienceBase (*Stanton et al., 2023*).

## RESULTS

### Detection patterns by release pool

In the GLMMs used to evaluate the telemetry receiver array network by pool we found neither detection frequency nor detection duration significantly differed (*p*-value < 0.05) between silver and bighead carp overall. There were however, some significant pool by species interactions (see Supplements 1 and 2). In a post-hoc comparison of pools, the detection frequency (number of individual dates with at least one detection) of silver carp tagged in the three lower reach pools (Alton, La Grange, and Peoria) were not significantly different from each other but had significantly lower detection frequency than the three upper reach pools (Starved Rock, Marseilles, and Dresden Island) except for Alton *vs.* Marseilles pools which were only nearly significant at $p = 0.05$. For bighead carp, only fish tagged in the La Grange and Peoria pools in the lower reach had significantly fewer detection dates than all three of the upper reach pools, while bighead carp tagged in the Alton pool were similar to the fish tagged Starved Rock pool (Table 1).

The detection duration (total length of time in days between first detection and last detection) of silver carp progressively increased from the downstream pools moving

**Table 2 Estimated marginal means of detection duration (number of days between first and last detection) and pair-wise comparisons between pools.** Numbers on the diagonal are estimated marginal means for pools after accounting for random year effects. Numbers above the diagonal are *p*-values from *post hoc* pairwise comparisons between pools (values < 0.05 in bold).

|  | Alton | La Grange | Peoria | Starved Rock | Marseilles | Dresden Island |
|---|---|---|---|---|---|---|
| **Silver carp** | | | | | | |
| Alton | [60.1] | 1.0000 | 0.0762 | **0.0072** | **0.0005** | **<0.0001** |
| La Grange | | [55.7] | **0.0002** | **<0.0001** | **<0.0001** | **<0.0001** |
| Peoria | | | [228.1] | 0.9881 | 0.2241 | **0.0050** |
| Starved Rock | | | | [286.9] | 0.6807 | 0.1142 |
| Marseilles | | | | | [379.4] | 0.9294 |
| Dresden Island | | | | | | [503.7] |
| **Bighead carp** | | | | | | |
| Alton | [125.5] | 1.0000 | 0.2707 | 0.8364 | **0.0036** | **<0.0001** |
| La Grange | | [140.6] | 0.1164 | 0.9430 | **0.0081** | **<0.0001** |
| Peoria | | | [30.6] | **0.0014** | **<0.0001** | **<0.0001** |
| Starved Rock | | | | [231.1] | **0.0258** | **0.0002** |
| Marseilles | | | | | [471.1] | 0.9073 |
| Dresden Island | | | | | | [643.0] |

upstream toward the Dresden Island pool. Silver carp tagged in the Alton and La Grange pools were not significantly different from each other but both pools had significantly shorter detection duration than the three upper reach pools. Detection duration for silver carp tagged in the Peoria pool did not differ from the Starved Rock or Marseilles pools but was significantly shorter than for fish tagged in the Dresden Island pool. Bighead carp had the shortest detection duration in the Peoria pool in the lower reach. Detection duration for bighead carp tagged in all three of the lower reach pools was significantly lower than the two most upstream pools (Marseilles and Dresden Island) but fish tagged in the Starved Rock pool in the upper reach were not significantly different than fish tagged in the Alton or La Grange pools in the lower reach (Table 2).

## Total pool-to-pool movements

To get a general understanding of the movement patterns of bigheaded carps in the IWW and to assess how much information was lost by processing into a monthly timestep, we summarized all daily upstream and downstream movements with and without first processing to a monthly timestep. An example is illustrated in Supplemental Information 3. We found 65.2% of silver carp were not detected outside of the pool they were tagged in. Of the fish that had made movements from the pool they were tagged in, 18.7% made movements only in a downstream direction, 1.1% made only upstream movements, and 15.0% made movements in both upstream and downstream directions. Summarizing detections to pool occupancy on a monthly timestep resulted in a greater percentage (76.8%) only occupying the pool they were tagged in, similar percentages making either downstream (19.3%) or upstream (1.4%) pool transitions, and a lower percentage (2.5%) showing both upstream and downstream pool movements, compared to examining all movements. Bighead carp showed a similar overall pattern with the majority of tagged

**Table 3 Summary of upstream/downstream movements for silver and bighead carp included in analysis after data cleaning to remove suspected dead fish and dropped tags.** Data are totals (percentages) of tagged fish over the movement history recoded from 2012–2019. Movements for 'All detections' are summarized over the entire detection history. Movements for 'Monthly summary' are summarized after categorizing data into monthly occupancies.

| *Silver carp; 353 total fish* | No movement | Downstream | Upstream | Both up/down |
|---|---|---|---|---|
| All detections | 230 (65.2) | 66 (18.7) | 4 (1.1) | 53 (15.0) |
| Monthly summary | 271 (76.8) | 68 (19.3) | 5 (1.4) | 9 (2.5) |
| *Bighead carp; 170 total fish* | | | | |
| All detections | 100 (58.8) | 27 (15.9) | 3 (1.8) | 40 (23.5) |
| Monthly summary | 117 (68.8) | 37 (21.8) | 2 (1.2) | 14 (8.2) |

fish only detected in the pool where they were tagged, but a greater percentage after summarizing data to a monthly timestep (58.8% *vs.* 68.8%). Downstream only movements (15.9% all detections; 21.8% monthly summary) were detected more than upstream only movements (1.8% all detections; 1.2% monthly summary), and more fish were detected with both upstream and downstream movements prior to summarizing to a monthly timestep (23.5% all detections; 8.2% monthly summary; Table 3).

## Multistate model

From the multistate movement model, the probability of a fish moving to a different pool on a monthly timestep is the product of the survival probability, probability of remaining battery life, and the pool-to-pool transition probability. We examined the overall probabilities of making upstream or downstream movements for each pool by summing the mean probabilities for moving to any pool upstream or downstream from the focal pool, respectively (Table 4; see Table S4 for total observations in each pool and Table S5 for posterior summary of all model parameters). We then examined the overall mean probabilities for all movements upstream or downstream and found similar values across both species' models. From the posterior distributions, the mean probability of remaining in the current pool was 0.76 (sd = 0.08; range across pools = 0.63–0.86) for silver carp and mean = 0.76 (sd = 0.04; range = 0.69–0.81) for bighead carp. The mean probability of movement to a pool upstream from any pool below Dresden Island was mean = 0.07 (sd = 0.08; range = 0.00–0.19) for silver carp and mean = 0.08 (sd = 0.06; range = 0.02–0.15) for bighead carp. The mean probability of movement to a downstream pool from any pool above Alton was mean = 0.14 (sd = 0.10; range = 0.01–0.27) for silver carp and mean = 0.14 (sd = 0.09; range = 0.02–0.22) for bighead carp.

Model estimates for survival and battery life were also similar across both species' models. Survival estimates from the posterior distributions were median = 0.963 (95% credible interval [CI] = 0.956–0.970) for silver carp and median = 0.969 (CI = 0.961–0.976) for bighead carp. The battery life parameter was median = 0.971 (CI = 0.967–0.974) for silver carp and median = 0.968 (CI = 0.963–0.972) for bighead carp.

To account for the variable numbers of receivers deployed through time in each pool (Fig. 2), we made detection probability a function of the number of receivers available each month within the models (see Methods). For both silver and bighead carp, we found

**Table 4 Mean monthly movement probabilities from the Bayesian multistate models for pool-to-pool movement in the Illinois River for silver carp (*Hypophthalmichthys molitrix*) and bighead carp (*H. nobilis*).** Movement probabilities are the product of survival, battery life, and pool transition probabilities. Numbers on the diagonal signify remaining in the same pool, numbers below the diagonal signify upstream movements, and numbers above the diagonal signify downstream movements. See Supplemental Information for estimates of all model parameter estimates with their 95% credible intervals.

| *Silver carp* | Alton | La Grange | Peoria | Starved Rock | Marseilles | Dresden Island |
|---|---|---|---|---|---|---|
| Alton | 0.747 | 0.007 | 0.000 | 0.000 | 0.139 | 0.132 |
| La Grange | 0.007 | 0.824 | 0.270 | 0.040 | 0.059 | 0.000 |
| Peoria | 0.000 | 0.050 | 0.626 | 0.039 | 0.002 | 0.000 |
| Starved Rock | 0.001 | 0.029 | 0.039 | 0.856 | 0.010 | 0.000 |
| Marseilles | 0.100 | 0.025 | 0.000 | 0.000 | 0.724 | 0.011 |
| Dresden Island | 0.080 | 0.000 | 0.000 | 0.000 | 0.001 | 0.792 |
| Total downstream | – | 0.007 | 0.270 | 0.079 | 0.210 | 0.143 |
| Total upstream | 0.188 | 0.104 | 0.039 | 0.000 | 0.001 | – |
| *Bighead Carp* | | | | | | |
| Alton | 0.783 | 0.074 | 0.001 | 0.001 | 0.060 | 0.102 |
| La Grange | 0.015 | 0.768 | 0.021 | 0.015 | 0.120 | 0.000 |
| Peoria | 0.001 | 0.008 | 0.806 | 0.206 | 0.000 | 0.057 |
| Starved Rock | 0.001 | 0.001 | 0.073 | 0.695 | 0.020 | 0.003 |
| Marseilles | 0.040 | 0.087 | 0.001 | 0.021 | 0.721 | 0.017 |
| Dresden Island | 0.097 | 0.000 | 0.036 | 0.000 | 0.016 | 0.758 |
| Total downstream | – | 0.074 | 0.022 | 0.222 | 0.200 | 0.179 |
| Total upstream | 0.154 | 0.096 | 0.110 | 0.021 | 0.016 | – |

that the model predicted higher detection probabilities, even at the lower range of receiver deployment, for the upper reach pools compared to the pools of the lower reach (Fig. 4). Over the duration of the study, detection probabilities for both silver and bighead carp were consistently high (mean >0.50) at the two most upper river pools, Marseilles and Dresden Island (Figs. 4B, 4D). Detection probability in Starved Rock pool increased to mean >0.50 early in the study (by early 2013) for both species and remained consistently high until the last months of the study period when receivers were removed. Detection probability varied for the Peoria pool over the duration of the study but was higher for silver than bighead carp. Probability of detection was consistently low (mean <0.25) for both species at Alton and La Grange for most of the duration of the study.

## DISCUSSION

### Movement patterns

The results of this study shed light on the movement patterns of silver and bighead carp species in the IWW and provide valuable insights that can assist in developing effective management and control strategies. Despite differences in the dataset used, data processing, and model construction, our findings on movement probabilities broadly align with a previous study of bigheaded carp movement in the IWW (*Coulter et al., 2018*). *Coulter et al. (2018)* found that most silver and bighead carp remain in their respective pools (mean = 0.83 for silver carp and mean = 0.81 for bighead carp) with higher probabilities for

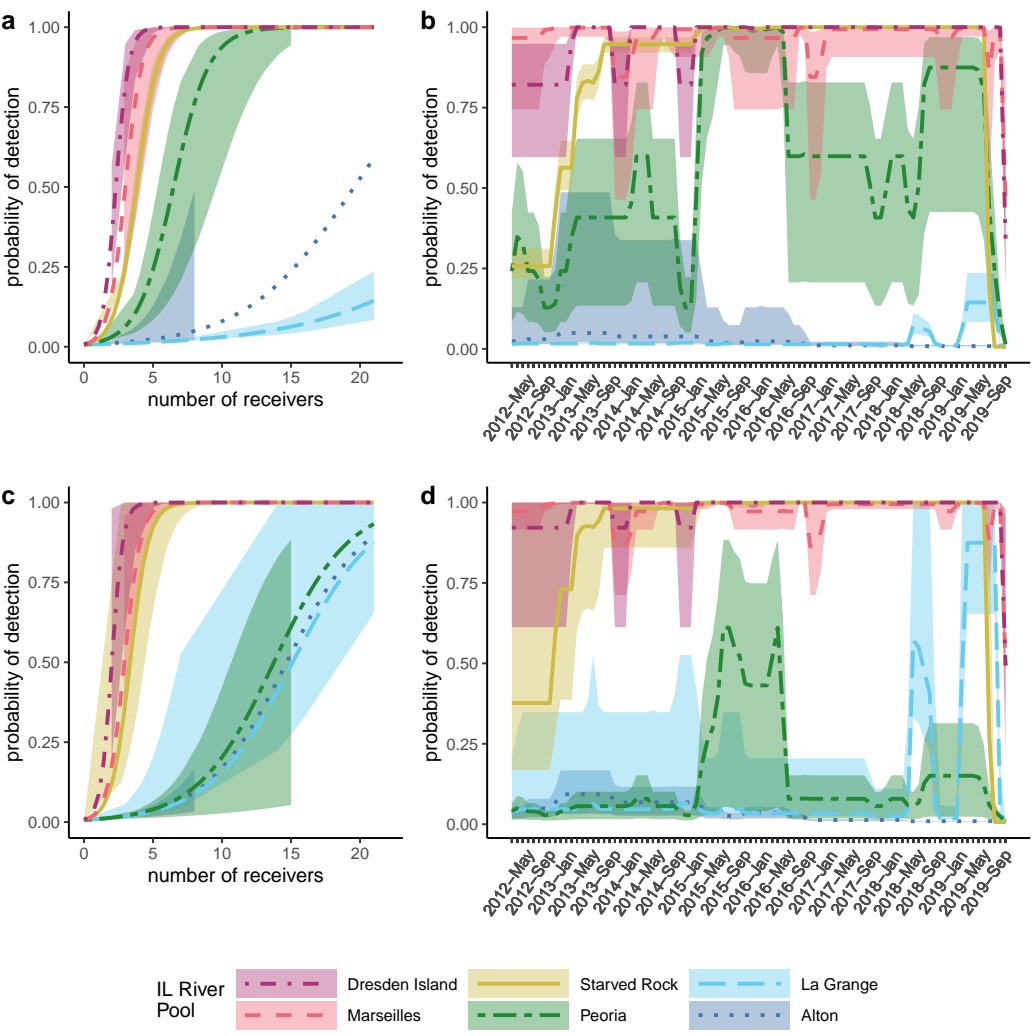

**Figure 4** **Modeled probabilities of detection by Illinois River pool.** For this study, Alton, La Grange, and Peoria pools are referred to as the 'lower reach', and Starved Rock, Marseilles, and Dresden Island pools compose the 'upper reach'. Solid lines show the median probabilities with shaded areas representing 95% credible intervals over the range of available data as a function of the number of receivers for (A) silver carp (*Hypophthalmichthys molitrix*) and (C) bighead carp (*H. nobilis*). Detection probability for each pool over the duration of the study based on the average number of receivers available for (B) silver carp and (D) bighead carp.

downstream movements over all pools (mean = 0.12 for silver carp and mean = 0.11 for bighead carp) than upstream movements (mean = 0.057 silver carp and mean = 0.095 for bighead carp). This consistency across species and studies indicates that these are robust patterns that can be used to generally predict the probability that bigheaded carps will make upstream or downstream movements.

However, our study differed in estimates at the level of individual pool-to-pool movements. For example, we found the highest probabilities of silver carp upstream movement from the lower river pools at Alton pool and La Grange pool (mean = 0.19

and mean = 0.10, respectively). In contrast, *Coulter et al. (2018)* found generally low levels of upstream movement from all lower reach pools (mean <0.05). Higher estimates of upstream movement from the lower reach pools would be consistent with the large numbers of bigheaded carps removed annually from the upper reach pools by the Illinois Department of Natural Resources (*MacNamara et al., 2016*; *ICRCC, 2023*) given the lack of evidence of successful recruitment in the upper river. If those populations are being maintained primarily through immigration from the lower pools where recruitment is occurring, higher estimated rates of upstream movement from the lower pools would be consistent with this observation. Nevertheless, given the low numbers of tagged fish in our dataset originating from the lower reach pools (Fig. 3) and the low detection probabilities estimated in those pools (Fig. 4) individual pool-to-pool movement estimates from our model may possibly be under or overestimated.

Previous studies have shown that bigheaded carps in the invaded waters of the United States are capable of making long-distance movements within short periods of time, with observed maximum movement rates as high as 95km/day (*DeGrandchamp, Garvey & Colombo, 2008*; *Coulter et al., 2016*). However, frequent sedentary behavior has also been observed (*DeGrandchamp, Garvey & Colombo, 2008*; *Coulter et al., 2016*; *Prechtel et al., 2018*; *Vallazza et al., 2021*; *Coulter, Prechtel & Goforth, 2022*). To accommodate the possibility of long-distance movements our model allowed for transitions between any pool within a monthly timestep, not just adjacent pools. However, using a monthly timestep resulted in the loss of some observed upstream and downstream movements when fish left and returned to same pool within a month (Table 3). These movements may potentially be associated with spawning activity (*Lubejko et al., 2017*) and warrant further investigation to enhance our understanding of carp staging behaviors and spawning movements in the IWW. Movement may also be motivated by intrinsic factors (*e.g.*, age, size, or past experiences in other pools) or extrinsic factors (*e.g.*, time of year, water temperature, water stage, dam structure, conspecific density, food resources) many of which have a seasonal component (*Coulter, Prechtel & Goforth, 2022*). These movements likely occur in short periods, less than our monthly timestep, and could be considered in future modeling efforts by using a shorter timestep possibly ranging from days to weeks.

### Telemetry array performance

Performance of the array depends on enough receivers being available where tagged fish are present to detect their transmissions. The array appeared to be more efficient closer to the invasion front at the three upper reach pools (*i.e.,* Starved Rock, Marseilles, and Dresden Island) relative to the lower reach (*i.e.,* Alton, La Grange, and Peoria). One way to measure this was by looking at the detection rates of tagged fish based on which pool they were tagged in. The detection frequency for fish tagged in the three upper reach pools was generally higher than fish tagged in the three lower reach pools for both species (Table 1). There was also a similar trend in the total time that tagged fish were detected (measured as the number of days between the first and last detections) showing a general decrease from upstream pools (estimated marginal mean = 504 days for silver carp and 643 days for bighead carp tagged in Dresden Island pool) to the most downstream pool (estimated

marginal mean = 60 days for silver carp and 126 days for bighead carp tagged in Alton pool; Table 2).

Receiver array performance is not the only possible explanation for differences in tagged fish detection patterns as other factors, such as those that would result in differential removal of tagged fish by pool (*e.g.*, permanent emigration, harvest, or mortality), could also lead to similar patterns. However, the evidence that the receiver array is more efficient at detecting fish in the upper reach pools is also supported by the detection probabilities from the multistate model. Detection probabilities were consistently high in the three upper reach pools over the duration of the study (Figs. 4B, 4D) with mean posteriors greater than 0.50, even at receiver numbers as low as 3–4 receivers per pool (Figs. 4A, 4C). Detection probabilities in the three lower reach pools were much lower with larger CIs, particularly in Alton and La Grange pools where mean detection probabilities from posterior distributions were less than 0.10 over most of the duration of the study for both species (Figs. 4B, 4D).

This discrepancy in array performance is likely driven by a combination of the habitat preferences of bigheaded carps coupled with inherent differences in habitat and habitat complexity between the upper reach pools and lower reach pools. Initial receiver placement was largely designed to identify pool to pool movements and so most receivers were located in the main channel. However, bigheaded carps tend to prefer backwater and side channel habitats (*DeGrandchamp, Garvey & Colombo, 2008*) and so may be less frequently detected in the main channel. Additionally, Dresden Island and Marseilles pools are smaller and offer comparatively fewer backwater and side channel habitats which may increase detections of bigheaded carps in the main channel of those pools. Adding receivers to backwater and side channel habitats would likely increase detections of fishes and add confidence to parameter estimates. However, backwater areas can be shallow in summer months which may limit receiver range in these locations. Receivers in the lower reach pools are also more vulnerable to loss due to the higher water volumes and shifting sediments which also contributed to reduced detections.

## Implications for future monitoring

Concentrating limited resources and effort to track fish movements closer to the invasion front is a logical strategy. However, the limited receiver coverage and few tagged fish relative to the size of the area in the lower reach pools compromises our ability to obtain an accurate understanding of movements at the scale of the river (*Gowan et al., 1994*; *Fausch et al., 2002*; *Kanno et al., 2020*). *Kallis et al. (2023)* also demonstrated the importance of taking a river-wide view of management for bigheaded carps with models that illustrate the effects that management actions (harvest and movement deterrents) taken in the lower reach pools have on the population sizes at the invasion front.

The lower reach pools are large and may benefit from more receivers or monitoring to achieve detection rates on par with the upper reach pools. We note that the dataset used in this study was constructed by pooling data from multiple agencies and multiple projects with varied goals and objectives. Receivers were frequently placed in the navigation channel of the river or in the vicinity of lock and dam structures to increase the likelihood

of detecting long-distance longitudinal movements or movements between pools (*ICRCC, 2012*; *ICRCC, 2015*) rather than increasing re-detections. However, the inability to re-detect fish can result in sparse datasets that can be difficult to fit when constructing multistate models. Increasing detection probabilities by installing additional stationary receivers, particularly in or near entrances to backwaters and side channels may also serve to increase detections. Closer analysis of detection success of individual receivers within pools with regards to the specific environmental conditions, mounting techniques, and locations relative to the main channel may also help to explain differences between the detectability of these two species across pools. It may also be possible to supplement stationary receivers with mobile surveys (*Coulter et al., 2016*; *Bunch et al., 2022*) in the lower reach pools. However, due to the size and complexity of the lower river, implementing mobile surveys could be costly and time consuming. Nonetheless, improving detection probabilities river-wide would enable exploration of additional models and co-variates that might have some explanatory power to enhance our understanding of fish movements. The compilation of this multi-agency dataset and conducting these analyses has already resulted in increased coordination across agencies to enhance tagging efforts and receiver placement in the lower reach pools.

In our multistate model we assumed a closed system and did not explicitly model emigration beyond these six pools on the Illinois River. This assumption is likely accurate for the upper reach pools because bigheaded carps are vigilantly monitored above Brandon Road Lock and Dam (*ICRCC, 2022*; Fig. 1). However, no barrier exists between Alton pool and the Mississippi River, and monitoring effort is currently sparse around this confluence. In our model, it is not possible to separate the rate that fish move downstream from Alton pool and do not return from mortality. Our model also does not estimate the rate at which fish enter the IWW from the Mississippi River which was previously estimated at 11–39% of silver carp but <3% of bighead carp (*Norman & Whitledge, 2015*). Not including sufficiently broad spatial sampling can bias models toward results that imply fish are more sedentary than they are (*Gowan et al., 1994*). This could have implications for management if the rate at which fish are moving into and out of this system is significant. If receiver coverage was increased in Alton pool and below the confluence with the Mississippi River, coupled with increased effort to tag fish in these areas, we could modify future iterations of the model to estimate movement into and out of the IWW.

## CONCLUSIONS

The ecological and economic damage that could result if the bigheaded carp invasion reaches the Great Lakes is potentially severe. This necessitates that investigations into the dynamics of the invasion proceed while new strategies are tested and while management actions are taken to contain the invasion front. Research is currently underway to further develop strategies to manage the bigheaded carp invasion efficiently and effectively. A diverse array of on-going research and investigation into new control methods (*Cupp et al., 2021*) is proceeding concurrently with the construction of deterrents, vigilant monitoring, and removal of bigheaded carp at the IWW invasion front (*ICRCC, 2022*; *ICRCC, 2023*).

Concentrating management actions primarily at the invasion front is unlikely to result in a satisfactory long-term control of bigheaded carp populations (*Garvey et al., 2015*; *Kallis et al., 2023*). There is currently a need for improved understanding of the movement dynamics and spatial ecology of bigheaded carps in this system for designing efficient management strategies (*Kallis et al., 2023*). Here we have provided updated movement estimates which suggest greater rates of movement from the lower reach toward the invasion front than previously estimated. However, there remains uncertainty in these estimates due to the unbalanced detection probabilities across river pools. This analysis provides a framework for evaluating the performance and efficiency of the telemetry array and can help inform decisions about the number and placement of telemetry receivers and tagging effort to meet future objectives for monitoring and management in the IWW. This framework can also be adapted for similar evaluations of invasive carps or other species in other river systems.

## ACKNOWLEDGEMENTS

The authors acknowledge the large number of biologists, students, and technicians from Southern Illinois University–Carbondale, US Army Corps of Engineers Chicago District, US Fish and Wildlife Service Carterville Fish and Wildlife Conservation Office Wilmington Substation, and Illinois Natural History Survey that spent many hours tagging fish and maintaining receivers in the collective telemetry array. We would like to acknowledge the Invasive Carp Regional Coordinating Committee's Monitoring and Response Workgroup (ICRCC MRWG) for continuous feedback throughout the project. We thank Tim J. Fox and Travis Harrison for their work on the Fishtracks database. This manuscript was improved by the thoughtful reviews of earlier drafts by Yu-Chun Kao, Hannah M. Thompson, and two anonymous reviewers. Any use of trade, firm, or product names is for descriptive purposes only and does not imply endorsement by the US Government.

### Funding

This work was supported through the U.S. Geological Survey Ecosystems Mission Area Biological Threats and Invasive Species Research Program and the Great Lakes Restoration Initiative (template number 774-M12 (USGS Telemetry Project: Real-time telemetry and multi-state modeling)) through the U.S. Environmental Protection Agency. The funders had no role in study design, data collection and analysis, decision to publish, or preparation of the manuscript.

### Grant Disclosures

The following grant information was disclosed by the authors:
The U.S. Geological Survey Ecosystems Mission Area Biological Threats and Invasive Species Research Program and the Great Lakes Restoration Initiative (template number 774-M12 (USGS Telemetry Project: Real-time telemetry and multi-state modeling)) through the U.S. Environmental Protection Agency.

## Competing Interests

The authors declare there are no competing interests.

## Author Contributions

- Jessica C Stanton conceived and designed the experiments, performed the experiments, analyzed the data, prepared figures and/or tables, authored or reviewed drafts of the article, and approved the final draft.
- Marybeth K Brey conceived and designed the experiments, authored or reviewed drafts of the article, and approved the final draft.
- Alison A Coulter conceived and designed the experiments, authored or reviewed drafts of the article, and approved the final draft.
- David R Stewart conceived and designed the experiments, authored or reviewed drafts of the article, and approved the final draft.
- Brent Knights conceived and designed the experiments, authored or reviewed drafts of the article, and approved the final draft.

## Data Availability

The data and code associated with this study are available on ScienceBase: Stanton, J.C., Brey, M.K., Knights, B.C., Coulter, A.A., and Stewart, D.R., 2023, Data to assess silver and bighead carp pool to pool movements from 2012 through 2019 in the Illinois River, USA through Bayesian multistate transition models (ver. 2.0, June 2024): U.S. Geological Survey data release, https://doi.org/10.5066/P9F82U46.

## Supplemental Information

Supplemental information for this article can be found online at http://dx.doi.org/10.7717/peerj.17834#supplemental-information.

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
