# Peer review of "Bayesian multistate models for measuring invasive carp movement and evaluating telemetry array performance"

_PeerJ, doi:10.7717/peerj.17834_

## Round 0.1 · original submission · Minor Revisions

Both reviewers appreciate the work you've put into the paper and have identified a few relatively small issues that will improve the paper (detailed comments from Reviewer 1 attached). Reviewer 2's main request was to provide the raw data (individual detections).

Reviewer 1 ·

Basic reporting

no comment

Experimental design

no comment

Validity of the findings

no comment

Additional comments

I've attached general and specific line comments to the review. Overall, the paper is very well written, but there are a few small issues with clarity in the writing, and some of the results.

Annotated reviews are not available for download in order to protect the identity of reviewers who chose to remain anonymous.

Reviewer 2 ·

Basic reporting

Generally, the manuscript is written very well. The authors used Bayesian multistate models to evaluate bigheaded carp movement and receiver detection performance. It is an innovative approach. As the authors mentioned, the manuscript has some limitations on movement results of the carps, for instance, the unbalanced receiver number and size between lower and upper river pools could result in bias of carp movement. Besides, the fish tag summary and code are available in (Stanton et al., 2023) but the receivers and their detections of the carps are not. I hope the detections of each tagged carp by each receiver can be available for everyone.

I have some suggestions to improve the manuscript:

1. Please add a citation for the selection of ‘Dirichlet priors’ in Line 317-19.

2. Please provide Gelman and Rubin’s convergence diagnostic in Line 323 to indicate the model converged or not.

3. Figure 4, in the title please add the abbreviation of ‘IL’ to Illinois, so that the ‘IL River Pool’ is reasonable in the legend.

Experimental design

no comment

Validity of the findings

no comment

Additional comments

no comment

---

## Round 0.2 · accepted · Accept

Thanks for submitting a revision -- the revised manuscript has addressed all of the authors suggestions, and is more clear throughout.